

# Prevalence of iron deficiency anemia in Brazilian women of childbearing age: a systematic review with meta-analysis

Mateus Macena[1,*], Dafiny Praxedes[1,*], Ana Debora De Oliveira[1],
Déborah Paula[1], Maykon Barros[1], André Silva Júnior[2], Witiane Araújo[1],
Isabele Pureza[2], Ingrid Sofia de Melo[3] and Nassib Bueno[1,2]

[1] Universidade Federal de Alagoas, Maceió, Brazil
[2] Universidade Federal de São Paulo, São Paulo, Brazil
[3] Instituto Federal de Alagoas, Satuba, Brazil
* These authors contributed equally to this work.

Corresponding author
Nassib Bueno,
nassib.bueno@fanut.ufal.br

## ABSTRACT

**Background:** Iron deficiency anemia (IDA) is among the most common micronutrient deficiencies in women of childbearing age and may affect children's development. Brazil has several national programs to tackle this condition, such as food fortification and supplementation for pregnant women, but IDA prevalence in this population has not been systematically reviewed. We sought to determine the prevalence of IDA in Brazilian women of childbearing age through a systematic review with metanalysis.

**Methodology:** A protocol was previously published on the PROSPERO platform under the code CRD42020200960. A panel of the National Council for Scientific and Technological Development (CNPq) approved the protocol of this study under the public call number 26/2019. The main databases searched were MEDLINE, Web of Science, Scopus, Lilacs, and SciELO. In gray literature, the Brazilian Digital Library of Theses and Dissertations and the annals of the Brazilian Congress of Epidemiology and the Brazilian Congress of Public Health were accessed. The search strategy involved terms related to the condition (IDA) and the age group of the population of interest (teenagers and adults). Studies that had assessed the prevalence of IDA in Brazilian women of childbearing age (10–49 years) were included. Three independent reviewers read all titles and abstracts and extracted data from the included studies. Random effects meta-analyses using the Freeman-Tukey arcsine transformation were carried out with prevalence data, and meta-regression was conducted to test for subgroup differences. The quality of the studies was assessed using the Newcastle-Ottawa Scale.

**Results:** From 21,210 unique records screened, 237 full-texts were retrieved, of which 91 were included in the qualitative synthesis, and 83 were included in the meta-analysis. The overall IDA prevalence was 25% (95% CI [23–28], 83 studies). The subgroup of studies that used random sampling showed a prevalence of 22% (95% CI [17–27], 22 studies), whereas in those with non-random sampling, the prevalence was 27% (95% CI [23–30], 61 studies), without significant differences between subgroups in the metaregression ($P = 0.13$). High prevalence of IDA were found in the subgroups of studies conducted in the North and Northeast regions (30%; 95% CI [24–37]; seven studies, and 30%; 95% CI [26–34]; 27 studies,

respectively), in studies conducted with indigenous population (53%; 95% CI [27–78], four studies), and with studies that had their collections after 2015 (28%; 95% CI [23–34], nine studies).

**Conclusions:** IDA in women of childbearing age remains a public health problem in Brazil, especially in the North and Northeast region. The national programs should be strengthened and more thoroughly supervised to decrease this condition nationally.

## INTRODUCTION

The current focus of the "1000-day window of opportunity", from conception to 2 years of life, has driven nutritional interventions to improve overall health of the maternal-child duo, but it often ignores the preconception period (*Dijkhuizen et al., 2019*). The high nutritional burden on women was recognized by the United Nations Sustainable Development Goals, aiming to meet the nutritional needs of adolescent girls, as well as pregnant and lactating women, by 2030 (*International Food Policy Research Institute, 2016*). In order to optimize pregnancy outcomes and avoid harmful effects on the growth, development and health of the fetus and child, it is clear that the nutritional and health status of women of childbearing age should be the starting point. Optimizing the status of micronutrients in this public may be a more effective and preventive strategy than targeted interventions only during pregnancy (*World Health Organization, 2009*).

Micronutrient deficiencies in women of childbearing age, in addition to harming women's health, affect pregnancy outcomes, delaying intrauterine growth and child development (*Grieger & Clifton, 2014*). Women are more vulnerable to micronutrient deficiencies due to their greater biological need and often due to the unequal distribution of food within the same household, by putting their families before their own needs (*Darton-Hill, 2012*). Among the most common micronutrient deficiencies in women is iron deficiency, which is the decrease in total body iron content (*Muthayya et al., 2013*). During their childbearing age, women are at increased risk of iron deficiency due to blood loss from menstruation and often have insufficient dietary iron intake to compensate for menstrual losses (*Simpson et al., 2011*). When the iron deficiency is severe, it may compromise the erythropoiesis process, leading to a decrease in hemoglobin concentration, a situation known as iron deficiency anemia (IDA), which affects a third of women of childbearing age worldwide (*Food and Agriculture Organization of The United Nations, 2017*). In Brazil, according to the National Survey on Demography and Health of Children and Women (PNDS) carried out in 2006, which was the last survey carried out with the objective of nationally evaluating women of childbearing age, the prevalence of IDA in this group was 29.4% (*Brazil, 2009*).

In order to tackle IDA, Brazil has several national programs, such as a food fortification program, which enriches wheat and corn flour with iron and folic acid, and was established in 2004 with the aim to decrease IDA (*Brazil, 2002*). In 2017, the Health Ministry of

Brazil updated the fortification requirements, which is now ranges from 4 mg to 9 mg of ferrous sulphate or fumarate per 100 g of flour (*Brazil, 2017*). A report from 2019 showed that only 61% of the flour samples analyzed showed adequate amounts of iron (*Brazil, 2020a*). Together with the national fortification program is the National Iron Supplementation Program, established in 2005, which aims to offer iron supplementation to children, pregnant women and women in the post-partum period (*Brazil, 2005*; *Brazil, 2013*). Hence, it would be expected that prevalence of IDA in Brazilian women of childbearing age should decrease throughout the years.

Therefore, collecting data on the iron status of Brazilian women of childbearing age is essential for the development of national public policies and for monitoring the effectiveness of existing programs. Thus, we aimed to determine, through a systematic review with meta-analysis, the prevalence of IDA in Brazilian women of childbearing age, and to analyze this prevalence in different contexts.

## MATERIALS AND METHODS

This systematic review with meta-analysis is reported according to Preferred Reporting Items for Systematic Reviews and Meta-Analyses (PRISMA) (*Page et al., 2021*). A protocol was previously published on the PROSPERO platform (http://www.crd.york.ac.uk/PROSPERO), under the code CRD42020200960.

The protocol of this study was approved by a panel of the National Council for Scientific and Technological Development (CNPq) under the public call number 26/2019. The presentation and defense of the project took place at the ground zero seminar on 04/17/2020. After that, the research was started and the submission to the registry of PROSPERO took place in sequence. Although in the registry of PROSPERO it is stated that on the date of registration we had already carried out the formal screening, this same protocol was already evaluated and authorized by the CNPq.

### Eligibility criteria

Observational, cross-sectional and/or prospective or retrospective cohort studies were included, dealing with the prevalence of IDA in women aged between 10 and 49 years, pregnant or not. For the cohort studies, only cross-sectional data at the end of follow-up that reported the prevalence of IDA were considered. Only studies carried out in Brazilian territory were included. When a study had more than one publication, the publication with the highest number of individuals evaluated was chosen.

### Information sources

Searches in MEDLINE, Web of Science (WOS), Scopus, Lilacs, SciELO databases were carried out until September 24, 2021. In gray literature, the Brazilian Digital Library of Theses and Dissertations (http://bdtd.ibict.br) and the annals of the Brazilian Congress of Epidemiology and the Brazilian Congress of Public Health, both available on the ABRASCO website (www.abrasco.org.br) was accessed to obtain studies developed in the Brazilian population that were not published in indexed media in the aforementioned conventional databases.

## Search strategy

For the construction of the search strategy, we used the acronym CoCoPop (condition, context and population). The search strategy involved terms related to the condition (IDA) and the age group of the population assessed (teenagers and adults). The context, which would include terms related to Brazil, was not used, as it limited the occurrences found in the databases. Two search strategies were used, one in English that had MeSH terms for MEDLINE and related free terms for Scopus and Web of Science, and another in Portuguese, using DeCS and Free Terms for LILACS and SciELO, in addition to the databases of gray literature. The following search was used in the MEDLINE database: (("iron"[MeSH Terms]) OR (iron deficiency anemia [MeSH Terms]) OR (ferritin [MeSH Terms])) AND ("prevalence"[MeSH Terms]) AND (("adolescent"[MeSH Terms]) OR ("adult"[MeSH Terms]) OR ("women"[MeSH Terms])).

## Data collection process

The study selection process consisted of two stages: the title and abstract reading phase and the full-text reading phase. All occurrences were read by two independent authors (MM and DP). In addition, five other authors (AO, DP, MB, WA and IP) each read 20% of the total occurrences. Hence, at least three independent authors read all reports. Disagreements were resolved from another judgment carried out by NB. The data collection process consisted of acquiring data derived from the eligibility criteria. For this purpose, an electronic spreadsheet was used to organize the outcomes and additional variables collected. When cases of absence of reported data were noticed, the authors of recent studies (2018–2021) included were contacted in order to obtain more information.

## Data items

The outcome evaluated was the prevalence of IDA. The complementary variables collected were the following elements: name of the study authors; region and state of study; year of study; type of study; age group; pregnant (yes/no); study context; method used to assess hemoglobin levels; and total number of subjects analysed.

## Effect measures

The prevalence of IDA was given as a percentage and was considered when hemoglobin levels were below 12 g/dL for non-pregnant women and below 11 g/dL for pregnant women (*World Health Organization, 2011*). When the prevalence of participants with IDA was not available, it was calculated from the absolute frequencies of individuals with IDA and the total sample evaluated.

## Quality assessment of included studies

To assess the methodological quality of the studies, a spreadsheet was built based on the Newcastle-Ottawa Scale, which is widely used for observational studies (*Zeng et al., 2015*), but adapted for cross-sectional studies. This scale gives "stars" to studies that meet

quality assumptions. In the present analysis, five categories were evaluated, considering that only prevalence studies were reviewed and that calculations of associations with risk measures will not be considered in this study.

The studies could obtain a maximum of 7 stars, as explained below: (a) representativeness of the chosen sample (2 stars in the case of representative samples of the population, with random sampling, 1 star in the case of non-random sampling, 0 stars in the case of absence of description for sampling); (b) adequate sample size (1 star for justified and satisfactory sample sizes, with sample size calculation, 0 stars for unjustified sample size); (c) assessment of non-respondents (1 star if there was an appreciation of non-respondents and indications that they do not differ from respondents; 0 stars if non-respondents were not mentioned, or that they are systematically different from respondents); (d) diagnostic criteria (1 star if you used the diagnostic criteria referenced in this protocol, 0 stars if you used other diagnostic criteria); (e) measurement of biochemical markers (2 star if there was a complete and adequate description of the measurement methods of biochemical markers, such as intra-individual coefficient of variation tests, 1 star if reporting an adequate method but without coefficient of variation, 0 stars if not there was a description of the method or if it was a method different from that provided for in this protocol).

## Overall evidence quality

The quality of the evidence was analyzed through adaptations of the method proposed by the Grading of Recommendations Assessment, Developing and Evaluation (GRADE) (*GRADE Working Group, 2017*). In the present analysis, this method was adapted for cross-sectional studies. The quality of evidence was classified in three categories: high, moderate and low, based on two criteria: limitations of the studies (quality assessment) and inconsistency of results (heterogeneity). Only these criteria were used due to the inadequacy of analyzing the traditional criteria "indirect evidence", "inaccuracy" and "publication bias" given the nature of the studies included.

## Data analysis

Data analysis was based on a quantitative study of the variables. Stata v.12 software (StataCorp, College Station, TX, USA) was used for this investigation, through the metaprop command (*Nyaga, Arbyn & Aerts, 2014*), with a DerSimonian and Laird random effects model using the Freeman-Tukey arcsine transformation to stabilize the variances. The data analyzed were the prevalence of IDA. Heterogeneity was assessed using the $I^2$ statistic, being considered high when the $I^2$ is greater than 50%. In addition, subgroup analyzes were performed by macro-geographic region of Brazil where the study was conducted, age group, pregnant women (yes/no), collection period and epidemiological context of collection. Finally, a meta-regression was also performed with the prevalence of IDA as the dependent variable and the score obtained by the Newcastle-Ottawa scale as the independent variable.
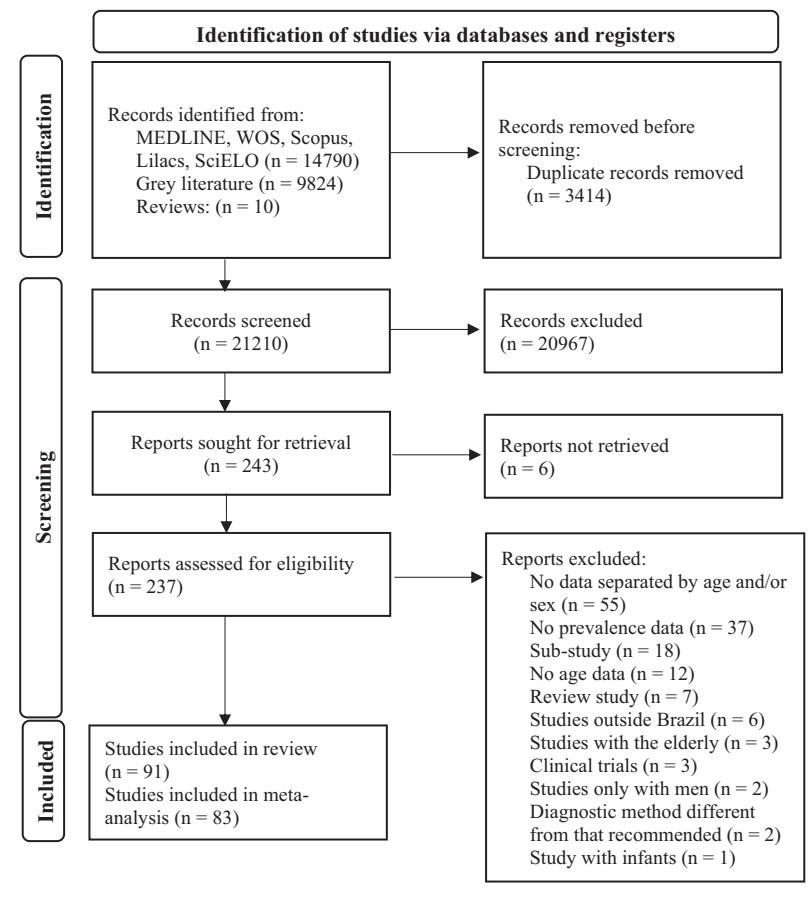

WOS: Web of Science

**Figure 1** **Flowchart of study selection.**

## RESULTS

### Study selection

At total, 24,624 occurrences were identified in the evaluated databases. Of these, 237 were selected for full-text reading. Finally, 91 studies were included for the qualitative synthesis and 83 for the meta-analysis (*Américo & Ferraz, 2011*; *Araf et al., 2010*; *Araújo, 2012*; *Araújo et al., 2013*; *Arruda, 1990*; *Arruda, 1997*; *Bagni, Luiz & Da Veiga, 2013*; *Batista Filho & Romani, 2002*; *Bezerra et al., 2018*; *Borges et al., 2016*; *Bresan et al., 2018*; *Bresani et al., 2007*; *Carvalho et al., 2017*; *Cavalcanti et al., 2014*; *Cavalcanti, Diniz & Arruda, 2019*; *Cintra, 2018*; *Clemente, 2019*; *Coelho, 2011*; *Côrtes, 2006*; *Da Costa et al., 2013*; *Da Silva, 2015*; *Dal Pizzol, Giugliani & Mengue, 2009*; *Dani et al., 2008*; *De Camargo et al., 2013*; *De Carli et al., 2018*; *De Castro et al., 2019*; *De França, 2006*; *De Oliveira, De Barros & Ferreira, 2015*; *De Sá et al., 2015*; *De Souza, 2011*; *Dell Agnolo, 2009*; *Demétrio, Teles-Santos & Dos Santos, 2017*; *Dos Santos et al., 2020*; *Dos Santos et al., 2018*; *Einloft et al., 2010*; *Fabian et al., 2007*; *Fávaro, 2011*; *Ferreira et al., 1998*; *Ferreira et al., 2007*; *Ferreira, Moura & Cabral Júnior, 2008*; *Ferreira, 2016*; *Frota, 2013*; *Fujimori, Szarfarc & De Oliveira, 1996*; *Fujimori et al., 1999*; *Fujimori et al., 2011*; *Guerra et al., 1990*; *Hirata et al., 2017*;

**Table 1 Summary of the characteristics of the included studies (*n* = 91).**

|  | N | % |
|---|---|---|
| **Age group** | | |
| Teenagers | 16 | 17.58 |
| Adults | 17 | 18.68 |
| Teenagers and adults | 48 | 52.75 |
| Not reported | 10 | 10.99 |
| **Pregnancy** | | |
| Yes | 53 | 58.24 |
| No | 31 | 34.07 |
| Both | 4 | 4.40 |
| Pregnant and non-pregnant separately | 3 | 3.29 |
| **Region** | | |
| North | 5 | 5.50 |
| Northeast | 26 | 28.57 |
| Midwest | 7 | 7.69 |
| South | 14 | 15.39 |
| Southeast | 34 | 37.36 |
| More than one | 2 | 2.20 |
| All | 3 | 3.29 |
| **Source** | | |
| Regular paper | 60 | 65.9 |
| Dissertation/theses | 21 | 23.1 |
| Congress abstract | 3 | 3.3 |
| Other sources | 7 | 7.7 |
| **Sampling** | | |
| Random | 24 | 26.4 |
| Non-random | 67 | 73.6 |
| **Context** | | |
| Household survey | 15 | 16.48 |
| Hospital | 15 | 16.48 |
| Outpatient/Basic health unit | 39 | 42.86 |
| School/University/Day care center/Swim club | 12 | 13.19 |
| Indigenous village | 7 | 7.69 |
| Hospital and Outpatient | 1 | 1.10 |
| Not reported | 2 | 2.20 |
| **Measurement method** | | |
| Hemocue | 24 | 26.37 |
| Cyanomethemoglobin | 15 | 16.48 |
| Cell-Dyn | 7 | 7.69 |
| Sysmex | 4 | 4.40 |
| Pentra | 3 | 3.30 |
| Agabe | 3 | 3.30 |
| BC 2800 e BS220 | 1 | 1.09 |
| Coulter | 1 | 1.09 |
| Cell Counter CELLM | 1 | 1.09 |
| Cyanide-free photometry | 1 | 1.09 |
| Medical record | 17 | 18.68 |
| Not reported | 14 | 15.39 |

*Instituto Nacional de Alimentação e Nutrição, 1998*; *Leite, 1998*; *Lerner, 1994*; *Lopes et al., 2006*; *Lucyk, 2006*; *Machado et al., 2016*; *Magalhães et al., 2018*; *Mariath et al., 2006*; *Marin et al., 2015*; *Marion, 2013*; *Marques et al., 2015*; *Massucheti, 2007*; *Miranda et al., 2018*; *Neves, 2018*; *Niquini et al., 2012*; *Orellana et al., 2011*; *Orsolin et al., 2020*; *Papa et al., 2003*; *Pereira, 1997*; *Pereira et al., 2019*; *Pessoa et al., 2015*; *Pincelli et al., 2018*; *Pinho-Pompeu et al., 2017*; *Quintans, 2011*; *Renz, 2018*; *Rezende, 2007*; *Rocha et al., 2005*; *Roncada & Szarfarc, 1975*; *Rondó & Tomkins, 1999*; *Sales et al., 2021*; *Santos, 2006*; *Santos et al., 2009*; *Santos et al., 2012*; *Saunders et al., 2016*; *Sena de Lira, 2009*; *Silla et al., 2013*; *Silva, Santos & Oliveira, 2018*; *Silva et al., 2020*; *Sinisterra Rodriguez, Szarfarc & Benicio, 1991*; *Szarfarc, 1974*; *Szarfarc et al., 1982*; *Szarfarc, 1985*; *Tapia et al., 2010*; *Walter et al., 2021*).

Details for the selection of studies can be seen in the flowchart in Fig. 1.

## Studies characteristics

Of the 91 studies included, 48 (52.75%) of them included samples with both age groups (adolescents and adults). Fifty-three (58.24%) studies were performed with samples from pregnant women. By the Brazilian macrogeographic division, most studies were carried out exclusively in the Southeast region ($n = 34$, 37.36%) and a minority exclusively in the North region ($n = 5$, 5.50%). The summary with the other characteristics of the included studies can be seen in Table 1. The characteristics of each study can be seen in Table S1.

## Quality assessment of included studies

The result regarding the assessment of the quality of the included studies can be seen in Table S2 and the sum of stars in this assessment is also shown in Table S1. The median score obtained by the studies was 3. The domains that presented the greatest inadequacies in the assessment according to the Newcastle Ottawa scale were related to the assessment of non-respondents ($n = 59$) and sample size ($n = 58$). The results of the meta-regression showed no significant association between the prevalence of iron deficiency and the score obtained by the Newcastle Ottawa scale ($\beta = -1.7\%$; 95% CI [$-0.038$ to $0.003$]; $p = 0.09$).

## Result of syntheses

The quantitative assessment of the 83 studies included in the meta-analysis can be seen in Table 2. Eight studies did not participate in the quantitative analyzes due to inconsistency in the data regarding the prevalence of IDA [29,30,37,51,53,58,85,104], which could interfere with our final results. The prevalence of IDA in all studies included in this analysis was 25% (95% CI [23–28]; $I^2 = 97.94$; 83 studies, 57,981 participants). The subgroup of studies that used random sampling showed a prevalence of 22% (95% CI [17–27], 22 studies) whereas in those with non-random sampling the prevalence was 27% (95% CI [23–30], 61 studies), without significant differences between subgroups in the meta-regression ($\beta = -5\%$; 95% CI [$-13.2$ to $1.7$]; $P = 0.13$).

Among the macro-geographic regions of Brazil, the North and Northeast regions showed the highest prevalence, 30% (95% CI [24–37]; 7 studies; 5,328 participants) and

**Table 2 Summary of the results found from the meta-analysis of the prevalence of iron deficiency in Brazilian adolescents and adult women of reproductive age (83 studies included).**

| | Studies | $n$ | $N$ | Pooled prevalence (%) | Lower 95% CI | Upper 95% CI | $I^2$ |
|---|---|---|---|---|---|---|---|
| Total | 83 | 15,075 | 57,981 | 25 | 23 | 28 | 97.94 |
| **Region** | | | | | | | |
| North | 7 | 1,752 | 5,328 | 30 | 24 | 37 | 95.35 |
| Northeast | 27 | 5,400 | 19,087 | 30 | 26 | 34 | 97.02 |
| Midwest | 7 | 518 | 2,177 | 24 | 16 | 33 | 95.09 |
| South | 15 | 2,343 | 11,231 | 20 | 14 | 27 | 98.59 |
| Southeast | 36 | 4,944 | 19,985 | 21 | 17 | 26 | 98.08 |
| **Age group** | | | | | | | |
| Teenagers | 16 | 839 | 3,580 | 18 | 11 | 27 | 97.44 |
| Adults | 16 | 2,211 | 8,091 | 26 | 21 | 31 | 95.93 |
| Both | 42 | 8,525 | 33,527 | 26 | 22 | 29 | 98.15 |
| **Pregnant women** | | | | | | | |
| Yes | 51 | 11,475 | 42,224 | 26 | 23 | 30 | 98.19 |
| No | 32 | 3,334 | 14,739 | 22 | 18 | 27 | 97.23 |
| Both | 3 | 266 | 1,018 | 42 | 17 | 69 | 96.86 |
| **Year collection** | | | | | | | |
| <2005 | 23 | 6,468 | 22,313 | 25 | 21 | 29 | 97.53 |
| 2005–2010 | 25 | 3,673 | 19,334 | 25 | 21 | 30 | 97.94 |
| 2010–2015 | 16 | 1,370 | 5,553 | 23 | 16 | 30 | 97.66 |
| ≥2015 | 9 | 2,543 | 7,291 | 28 | 23 | 34 | 95.18 |
| **Context** | | | | | | | |
| Household survey | 13 | 1,954 | 8,130 | 24 | 20 | 39 | 93.73 |
| Hospital | 13 | 2,757 | 7,753 | 36 | 31 | 40 | 92.83 |
| Outpatient/Basic health unit | 40 | 8,486 | 35,562 | 21 | 18 | 24 | 98.38 |
| School/University/Day care center/Swim club | 12 | 527 | 2,309 | 25 | 14 | 37 | 97.71 |
| Indigenous village | 4 | 305 | 923 | 53 | 27 | 78 | 97.65 |
| **Sampling method** | | | | | | | |
| Random | 22 | 2,169 | 10,059 | 22 | 17 | 27 | 96.95 |
| Non-random | 61 | 8,795 | 30,646 | 27 | 23 | 30 | 97.99 |

**Table 3 Evidence quality assessment.**

**Question: What is the prevalence of iron deficiency in Brazilian women of childbearing age?**

| Quality assessment | | | | | | Results summary | Quality | Importance |
|---|---|---|---|---|---|---|---|---|
| No of studies | Study design | Risk of bias | Inconsistency | Indirect evidence | Imprecision | Other considerations | Pooled prevalence (%) | | |
| Prevalence of iron deficiency (rated with: %) | | | | | | | | | |
| 83 | Observational studies | Serious | Very serious | N/A | N/A | None | 25% (95% CI [23–28]) | ⊕○○○ Very Low | IMPORTANT |

30% (95% CI [26–34]; 27 studies; 19,087 participants), respectively. Meta-regression analysis revealed that macro-geographic region was a significant predictor of the IDA prevalence ($\beta$ = −2.8%; 95% CI [−5.1 to −0.4]; $P$ = 0.01).

Other subgroups that showed high prevalence of IDA were those with studies with only adult women (26%; 95% CI [21–31]; 16 studies; 8,091 participants); which included samples of pregnant and non-pregnant women together (42%; 95% CI [17–69]; 3 studies; 1,018 participants); with indigenous people (53%; 95% CI [27–78]; 4 studies; 923 participants); conducted in the hospital context (36%; 95% CI [31–40]; 13 studies; 7,753 participants); and studies that had their collections after 2015 (28%; 95% CI [23–34]; 9 studies; 7291 participants).

### Certainty of evidence

Considering the limitations of the studies included in this review and the inconsistency of the results, the quality of evidence was considered very low (Table 3).

## DISCUSSION

This is the first systematic review to pool the prevalence of IDA in Brazilian women of childbearing age. IDA is estimated to affect about a third of women of childbearing age in the world (*World Health Organization, 2021a*). The regions of the globe in which this prevalence is higher are Sub-Saharan Africa, South Asia, the Caribbean and Oceania (*Kassebaum et al., 2014*). Despite not being part of the regions mentioned, the results of the present study indicate a prevalence of 25% for this condition in Brazil. Such prevalence is considered high, and it was even higher among women in the North and Northeast regions of the country. Furthermore, it is noteworthy that overall, the quality of the included studies was low. To overcome this limitation, we conducted subgroup analyses and meta-regression analysis. Nevertheless, the prevalence of IDA did not show significant differences between studies that used a random sampling approach *vs* those with non-random sampling.

According to the World Health Organization (WHO), the prevalence of IDA is considered a public health problem when it is above 5%, which gives a degree of significance to the respective severity of this prevalence in the present study (*World Health Organization, 2011*). Thus, all the grouped prevalence of IDA found in this review fit the classification established by the WHO, ranging from a problem of mild significance (18% of adolescents) to severe (53% of indigenous women). In addition, our findings show a higher prevalence than indicated by the WHO survey for Brazil in 2019 (16.1% for non-pregnant women and 19.1% for pregnant women) (*World Health Organization, 2021b*, *2021c*).

*Petry et al. (2016)* aimed to carry out, through a systematic review, a survey of the prevalence of IDA in women of childbearing age in countries with low, medium and high human development index. Countries such as Bangladesh (2011/2012), Cameroon (2012), Côte d'Ivoire (2007), Mongolia (2010) and Iraq (2011) showed prevalence (18.5–28.6%) similar to that found in our meta-analysis [115]. We are not aware of a systematic review that has assessed this prevalence in women of childbearing age in Brazil.

However, this type of investigation has already been carried out in children under the age of five. Systematic reviews with meta-analyses by *Silveira et al. (2021)* and *Ferreira et al. (2021)* show that compared to older studies, the prevalence of IDA in children under five years has been decreasing (*Ferreira et al., 2021*; *Silveira et al., 2021*). Reinforcing this, according to the Ministry of Health of Brazil, referring to the National Study on Food and Child Nutrition, the prevalence of anemia in Brazilian children has reduced by half in the last 13 years (*Brazil, 2020b*). These notes are different from what seems to be happening with women of childbearing age when comparing the studies by the data collection period in this meta-analysis.

As for the country's internal differences, according to data from the National Health Survey conducted between 2013 and 2014 with women aged over 18 years, the macro-regions of Brazil that had the lowest prevalence of iron deficiency anemia were the southern regions (9.0%) and Midwest (7.9%) (*Machado et al., 2019*). In our study with women of childbearing age, the lowest prevalence was found in the South (20%) and Southeast (21%) regions. However, in relation to the highest prevalence, both the National Health Survey (2013–2014) and our analyzes highlight the North (14.6% *vs* 30%, respectively) and Northeast (15.7% *vs* 30%, respectively) regions (*Machado et al., 2019*).

Such differences between the macro-regions within Brazil can be explained by the evident social inequality and distinct development between the macro-geographic regions, making the South-Southeast two regions with greater development, providing a better quality of life, and the opposite generates higher prevalence in the North-Northeast regions (*Ferreira Filho & Horridge, 2006*). The root of this inequality among Brazilian macro-regions is complex and involves several factors. Briefly, the South and Southeast regions of Brazil have historically received more incentives from Brazilian governments, especially regarding industrialization and allocation of foreigner's investments, which contributed to the social development of these regions. On the other hand, the North and Northeast region did not go through an industrialization process and remained with economic activities with lower aggregated value (*Monteiro-Neto, 2014*).

Paying attention to risk factors for IDA implies improving the quality of life of women of childbearing age, since this nutritional deficiency has already been shown to be associated with cognitive impairment and neuropsychiatric disorders increasingly common in children and adolescents, such as autism and anxiety disorder (*Islam et al., 2018*). In adolescence, it has an impact on physical and mental development, such as cognitive impairment, impairment of the proper functioning of the respiratory system, causing fatigue, fatigue and weakness, which can be considered a trigger for weight gain due to the lack of willingness to performing physical exercise (*Garzon et al., 2020*; *World Health Organization, 2001*). Furthermore, pregnant women who start this physiological period with IDA have higher risks of maternal and perinatal death, premature birth, low birth weight and infant morbidity (*Batista Filho, De Souza & Bresani, 2008*; *Lício, Fávaro & Chaves, 2016*), in addition to impacts during pregnancy itself, such as emotional instability, pre-eclampsia, cardiovascular and immune function changes, and lower physical and mental performance (*Garzon et al., 2020*).

Given the relevance of this public health problem to the lives of individuals, the Brazilian government implemented in 2005 the National Iron Supplementation Program, which consists of: prophylactic iron supplementation for specific population groups (for all children aged 0–6 months age, pregnant women starting prenatal care until the third month postpartum); fortification of food for children with powdered micronutrients; promotion of adequate and healthy diet to increase consumption of iron-rich foods; and the mandatory fortification of corn and wheat flour with iron and folic acid (*Brazil, 2013*). The studies conducted by *Araújo et al. (2013)* and *Fujimori et al. (2011)* indirectly showed that such flour fortification was able to reduce the prevalence of iron deficiency anemia in pregnant women (*Araújo et al., 2013*; *Fujimori et al., 2011*). However, we have not identified any studies that have evaluated this strategy in another population group for the interest of this systematic review.

This systematic review has some limitations that may influence our conclusions. First, we highlight the wide variation between the sizes and quality of the included studies, which possibly contributed to the finding of the high heterogeneity identified in the meta-analyses, as well as the very low quality of evidence. Still, we tried to explore *via* subgroup and meta-regression analysis the impact of the study quality in our results. In addition, the discrepancy in the number of studies between Brazilian macro-geographic regions is highlighted, which can make the comparison between them uneven. We also highlight the small number of studies aiming at evaluating indigenous populations, which may have made the pooled prevalence of this sample unrealistic in our analysis. Finally, we highlight the need for further studies to update the assessment of the prevalence of IDA in women of childbearing age, since few studies were conducted after 2015 and it was difficult to conduct a time-series comparison to highlight possible temporal changes in the prevalence of this nutritional deficiency.

## CONCLUSIONS

IDA in women of childbearing age remains a public health problem in Brazil, and the population in the North and Northeast of the country need greater care for the prevention and treatment of this condition. Thus, greater efforts by the Brazilian government are needed to strengthen and better supervise the national programs of iron supplementation and fortification.

### Funding

This study was funded by the National Council for Scientific and Technological Development–CNPq, with support from Decit/SCTIE/Ministry of Health of Brazil–MoH, from the General Coordination of Food and Nutrition of the Health Promotion Department of the Primary Health Care Secretariat of the Ministry of Health (CGAN/DEPROS/SAPS/MS), through the call no. 26/2019, with protocol number 442859/2019-8. The funders had no role in study design, data collection and analysis, decision to publish, or preparation of the manuscript.

## Grant Disclosures

The following grant information was disclosed by the authors:
National Council for Scientific and Technological Development–CNPq.
Decit/SCTIE/Ministry of Health of Brazil–MoH.
General Coordination of Food and Nutrition of the Health Promotion Department of the
Primary Health Care Secretariat of the Ministry of Health (CGAN/DEPROS/SAPS/MS):
26/2019 and 442859/2019-8.

## Competing Interests

The authors declare that they have no competing interests.

## Author Contributions

- Mateus Macena performed the experiments, analyzed the data, prepared figures and/or tables, authored or reviewed drafts of the paper, and approved the final draft.
- Dafiny Praxedes performed the experiments, analyzed the data, prepared figures and/or tables, authored or reviewed drafts of the paper, and approved the final draft.
- Ana Debora De Oliveira performed the experiments, prepared figures and/or tables, and approved the final draft.
- Déborah Paula performed the experiments, prepared figures and/or tables, and approved the final draft.
- Maykon Barros performed the experiments, prepared figures and/or tables, and approved the final draft.
- André Silva Júnior analyzed the data, prepared figures and/or tables, authored or reviewed drafts of the paper, and approved the final draft.
- Witiane Araújo performed the experiments, prepared figures and/or tables, and approved the final draft.
- Isabele Pureza performed the experiments, prepared figures and/or tables, and approved the final draft.
- Ingrid Sofia de Melo conceived and designed the experiments, analyzed the data, authored or reviewed drafts of the paper, and approved the final draft.
- Nassib Bueno conceived and designed the experiments, performed the experiments, analyzed the data, authored or reviewed drafts of the paper, and approved the final draft.

## Data Availability

The information from each study included in the systematic review is available in the Supplemental Tables.

## Supplemental Information

Supplemental information for this article can be found online at http://dx.doi.org/10.7717/peerj.12959#supplemental-information.

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
