# Peer review of "Prevalence of iron deficiency anemia in Brazilian women of childbearing age: a systematic review with meta-analysis"

_PeerJ, doi:10.7717/peerj.12959_

## Round 0.1 · original submission · Major Revisions

From the abstract, please remove the protocol information. Mention the key academic databases, search terms, and search methods. Along with the search results.

Reviewer 1 ·

Basic reporting

This article reviews iron deficiency anemia (IDA). The authors acknowledge that Brazil has several programs nationwide to deal with this condition in women. The authors propose a systematic review of the prevalence of iron deficiency anemia (IDA) in Brazilian women of childbearing age, as this condition affects the development of the fetus. For this, Science, Scopus, Lilacs, SciELO and gray literature databases were used. It was concluded that IDA in women of childbearing age remains a public health problem in Brazil.

Experimental design

Although an important survey was carried out in an adequate database, there are still some doubts:
1- the number was not clear, among the 83 works selected for meta-analyses, how many are gray literature;
2- another issue to be considered is the authors' choice of method. Meta-analysis relies on statistics that combine the results of various scientific studies to derive a pooled estimate closest to the truth and that identify patterns or sources of disagreement between the results. However, the authors show a paradox in table
3. This study is registered as important and at the same time it is alleged that there is a serious risk of bias, a very serious inconsistency and very low quality. It is suggested that this paradox be circumvented in the statistical (quantitative) methodology so as not to harm the research based on meta-analysis.

Validity of the findings

4- it is not clear what the innovative result of the research is, since the lack of iron in women is already a known factor and there are measures taken by the Brazilian government to overcome it. It is also known that in less wealthy regions (in the case of this article, northern and northeastern regions of Brazil) people have a low intake of necessary nutrients for health, including iron;
5- acronyms used in lines 148, 149 and 150 need to be specified the first time they are cited

Additional comments

It is suggested that the authors review the article and add a topic about the measures they should take so that the paradox does not occur. In this way the article will be improved to pass.

Annotated reviews are not available for download in order to protect the identity of reviewers who chose to remain anonymous.

Reviewer 2 ·

Basic reporting

I found it a nicely written paper with recent literature. The article is written in a scientific structure.

Experimental design

It is a novel idea to test. The research question and the problem are well defined. Databases were searched extensively.

Validity of the findings

It is an impactful work and implementation of results that would benefit the community people.

Additional comments

Well done. Thanks and congratulations.

·

Basic reporting

The article is well written and with the necessary language skills. The article addresses an important public health problem regarding the health conditions of a vulnerable population segment, such as women of reproductive age. The references used are sufficient and updated to some extent. Revisions of some used references are necessary, as they are outdated. The tables were well developed and are able to represent the data found well. The supplementary files were necessary and able to resolve doubts that arose during the reading and revision. The flowchart is sufficient and self-explanatory, demonstrating the entire process of searching and selecting the articles that make up the review. The article is explanatory, but needs some revisions for its enhancement and publication.

Some comments below emerged as the reading of the work progressed and need further consideration by the authors:

C1: The systematic review itself can not determine the prevalence of IDA in brazilian women. Include the meta-analyses in abstract's background.

C2: In introduction, it is necessary to explain the difference between iron deficiency itself and iron deficiency anemia, as the authors mention both as women's health problems.

C3: On line 91, the authors mention the high prevalence of IDA found in northeastern Brazil. It is important to explain to non-brazilians the differences between Brazil's geographic regions and its particularities, specially the northeast.

C4: On line 123 the authors mention that studies that evaluated iron deficiency were included in review' formal screening, but on line 125, iron deficiency anemia was mentioned. Which one was screened?

C5: In discussion, which conditions make the South-Southeast regions provide a better quality of life that North-Northeast can not attend?

C6: Without a proper meta-regression between the pooled prevalence and the sub-groups, the authors can not claim that some prevalences were higher than others.

Experimental design

The authors well defined their object of study and carried out an extensive literature search of papers eligible for meta-analysis. However, some issues were not very clear and raise doubts about the methodological rigor of the study.

C1: Were only studies with a probabilistic sample included in the meta-analysis? It is not clear in the article. If so, it is important to emphasize as a strength of the study and the quality of evidence found. If not, I suggest that they carry out the meta-analysis only with data from investigations with probability sampling, as convenience samples can bias the results found because they do not adequately represent the contingent population of women of reproductive age with iron deficiency anemia.

C2: I mentioned it before, but it is important to emphasize that the analysis of subgroups only aggregates the results found according to interest groups, but it cannot be affirmed, without an appropriate robust meta-regression analysis, that the prevalences found in a subgroup is superior to others.

Validity of the findings

The work is unprecedented, as there is no systematic review with meta-analysis of iron deficiency anemia in women of reproductive age in Brazil. Their findings are consistent with parallel investigations and data projections from international entities. However, the low quality of the findings, possibly resulting from the inclusion of studies without random sampling, may slightly bias their findings. The conclusions are consistent and respond to what was intended to be investigated.

Additional comments

No comment.

---

## Round 0.2 · accepted · Accept

Thanks for making all the necessary changes and providing a point by point explanation.

Reviewer 1 ·

Basic reporting

Please, see topic number 4

Experimental design

Please, see topic number 4

Validity of the findings

Please, see topic number 4

Additional comments

After verifying the insertions in the amended text, the topics listed by this reviewer on the occasion of the first review were clarified. This gave greater consistency to the content developed by the authors. Therefore, I consider the re-review made by the authors satisfactory and recommend the publication of the second version of the article.

·

Basic reporting

The authors made the necessary corrections, as well as informed when it was not possible to change some aspects. The work is now in a suitable format for publication, as well as adding great scientific value to the field of epidemiology of deficiency diseases in a vulnerable Brazilian segment.

Experimental design

No comment.

Validity of the findings

No comment

Additional comments

No comment.